# Research and Optimization of Process Parameters for Internal Thread Forming Based on Numerical Simulation and Experimental Analysis

**DOI:** 10.3390/ma15093160

**Published:** 2022-04-27

**Authors:** Qiang He, Yuxiang Jiang, Xuwen Jing, Yonggang Jiang, Honggen Zhou, Bofeng Fu

**Affiliations:** 1School of Mechanical Engineering, Jiangsu University of Science and Technology, Zhenjiang 212000, China; 202020032@stu.just.edu.cn (Y.J.); 209020062@stu.just.edu.cn (X.J.); 211210201311@stu.just.edu.cn (H.Z.); 2CSSC Huangpu Wenchong Shipbuilding Company Limited, Guangzhou 510715, China; 211110201214@stu.just.edu.cn; 3Shanxi Diesel Engine Heavy Industry Company Limited, Xi’an 713100, China; 211210201120@stu.just.edu.cn

**Keywords:** internal thread, thread forming, numerical simulation, physical test, parameter optimization

## Abstract

In order to improve the forming quality of extruded thread, finite element analysis and experimental research are combined to reduce the two keys that affect thread quality in the machining process—extrusion torque and extrusion temperature. The effects of different processing parameters on the extrusion torque and temperature are obtained by numerical simulation, including the bottom hole diameter of the workpiece, the machine tool speed, and the lubrication medium. For the purpose of reducing extrusion torque and temperature, the process parameters for internal thread forming are further optimized by orthogonal design. It is determined that when machining the M22 × 2 internal thread on the connecting rod of the marine diesel engine made of 42CrMo4 steel, the bottom hole diameter of the workpiece should be ∅21.20 mm, the speed of the machine tool should be 40 RPM, and the lubricating medium should be PDMS polydimethylsiloxane coolant. Compared to before optimization, the maximum extrusion torque and the maximum extrusion temperature are reduced by 19.27% and 15.07%, respectively. On the premise of ensuring the thread connection strength, the height of the thread tooth is reduced by 0.052 mm, and the surface condition of the thread is improved. The surface microhardness at the root, top, and side of the thread increases by about 5 HV_0.2_, and the depth of the hardened layer increases by 0.05 mm. The results show that the quality of the optimized thread is higher.

## 1. Introduction

With the rise of our country’s shipbuilding industry, the manufacturing requirements of machinery and equipment are developing in the direction of high strength, high precision, and long life. Threaded parts are a very important structural part and fastening connection in mechanical equipment, which can realize the connection and fixation between the parts [1]. According to statistics, more than 60% of the fastening connections are threaded connections, which can bear large radial, axial, and shear loads when the mechanical equipment is running; so, thread performance is directly related to the service life of aircraft, high-speed trains, ships, and so on [2,3]. Therefore, the designed thread parts need to meet the following characteristics: reliable performance, long service life, simple structure, and lightweight; this introduces higher and higher requirements for the machining of thread parts.

As a traditional thread machining method, many scholars have studied the thread cutting technology. Ma et al. [4] studied the dynamic problems in the tapping process by establishing a coupling model between transverse vibration, axial vibration, and dynamic cutting force on the tapping path and proved that the influence of transverse vibration and axial vibration on dynamic cutting force is decoupled. Faur et al. [5,6] reduced the production of chips during tapping by modifying the geometry of the spiral groove of the cutting tap, thus reducing the cutting torque and extending the service life of the tap. Matsui et al. [7] calculated the radial force of the thread through a four-component piezoelectric dynamometer and tapped the thread by the method of screw interpolation motion of the screw machine, so as to improve the accuracy of the formed thread. Monka et al. [8] monitored the wear of taps during cutting through an online vibration detection system, and the relationship between the helix angle and the forming quality of screw thread was studied. In order to reduce the torque during tapping and prolong the service life of tap, Golovkin et al. [9,10,11] introduced the ultrasonic wave into the cutting process. Miroslav et al. [12] studied the relationship between cutting temperature, cutting efficiency, thread quality, and material properties of the workpiece through experiments. In order to improve the torsional stiffness of tap and improve the cutting efficiency, Yin et al. [13] introduced vibration in the tapping process. Piska et al. [14,15,16,17] increased the cutting performance and reduced the wear of tap by coating the surface of the tap. Popovi et al. [18] analyzed the geometric structure of the cutting tool and established the prediction model of the tapping force and torque by defining the direction matrix of tap coordinate system; they then verified the accuracy of the model through experiments. Tanaka et al. [19] designed a method to measure the blade temperature in the tapping process based on a two-color pyrometer with optical fiber, and then used this method to study the relationship between the workpiece material and cutting temperature. Xu et al. [20,21] proposed a tap wear monitoring and prediction model system based on deep learning, then collected vibration signals and tap wear through a tapping test to verify the accuracy of the system.

Considering that the internal thread cut on the marine diesel engine connecting rod of 42CrMo4 steel may not meet the quality requirements of the enterprise, it is necessary to replace the cutting with cold extrusion, which has higher material utilization and better machining quality [22,23,24,25]. However, most of current research results on extruded thread are for specified metal and specific specification thread. When the processing environment such as tap structure, material property, type, and specification of thread changes, the forming result will also change [26,27,28,29]. For the M22 × 2 internal thread of 42CrMo4 steel, the existing processing parameters of thread in enterprises mostly rely on the production experience of workers, which means a lack of theoretical and experimental guidance. Therefore, the influence of process parameters on thread forming is analyzed by the method of numerical simulation and experiment, and the best processing parameters are selected.

In this work, the forming mechanism of the internal thread is first studied by the method of theoretical analysis. The changes of torque and temperature in the thread forming process are measured by experiment. The morphology, microstructure, and hardness of the thread are analyzed by means of the shape measurement laser microscope system and automatic micro-Vickers hardness measurement system. Then, the thread forming process is numerically simulated by finite element software DEEORM-3D. The effects of bottom hole diameter, machine tool speed, and lubricating medium on extrusion torque and temperature are studied by setting different parameters in the numerical simulation. In order to reduce the torque and temperature in the extrusion process, the processing parameters are optimized. Finally, the extrusion experiment is carried out with the optimized parameters and the extrusion torque and temperature before and after optimization are compared. The tooth height, surface microhardness, and hardened layer of the thread before and after optimization are also measured to evaluate the quality of thread.

## 2. Forming Mechanism and Experiment of Extrusion Thread

### 2.1. Forming Mechanism of Extruded Thread

The essence of internal thread forming is the process in which the plastic deformation of the workpiece occurs under the action of the extrusion tap and finally forms the thread. Figure 1a is a schematic diagram of the workpiece machined by an extruded tap. When the extrusion tap is spun into the workpiece at a certain speed, the inner wall of the workpiece is in contact with the first tooth, A, of the extrusion cone. Because the hardness of the tap is much higher than that of the workpiece, tooth A will extrude a shallow dent on the workpiece surface. The metal of the workpiece is plastically deformed by the extrusion of tooth A, which flows along both sides of tooth A and forms a bulge on both sides of the top of it. As the tap rotates once during the working process, the tap moves forward by a distance of pitch, and tooth B enters the indentation extruded by tooth A. Because there is a certain cone angle in the extrusion cone of the tap, the depth of tooth b is deeper than that of tooth A. On the basis of tooth A squeezing into the dent, tooth B further deepens the depth of the dent, increases the degree of plastic deformation of the workpiece, and increases the height of the bulge formed on both sides of tooth B’s tip. At this time, the metal between teeth A and B will show protrusion on both sides and flatness in the middle. With tooth C being squeezed into the workpiece, the dent left by tooth B is further extruded, and the amount of metal material flowing into the groove between teeth B and further increases. At this time, there will be a preliminary outline of the thread between teeth B and C. As the last tooth, D, of the extrusion cone is extruded into the workpiece, the degree of plastic deformation of the workpiece further increases, and the amount of metal material extruded between the two teeth further increases. At this point, a thread with high integrity and that is close to the qualified size will be formed between teeth C and D. The forming process of the thread under the action of the tap extrusion taper is shown in Figure 1b.

As the extrusion tap continues to move forward, the calibration part of the tap begins to extrude the workpiece. There is no cone angle in the calibration part. The function of the calibration part is to modify the shape and size of the thread formed by the extrusion cone. Until the workpiece is completely extruded by the edges of calibration part, a thread that meets the size requirements will be formed on the workpiece surface. The extruded thread is shown in Figure 1c. Because the flow velocity of metal on both sides of the tooth top is greater than that in middle, the thread after extrusion will be higher on both sides of the tooth top and concave in middle. This phenomenon is called lack of meat. It is inevitable that there will be more or less insufficiency of meat in the top part of the thread machined by the extruding tap. In addition, the lines in figure are represented as metal fiber tissue. The metal of the workpiece flows from the tooth root to the tooth top under the extrusion of the tap, and the surface fiber structure of metal is still continuous. Among the three parts of the thread, the plastic deformation range of the metal material at the tooth root is the largest, so the distribution of metal fiber is densest. The plastic deformation degree of metal at the tooth top is the smallest, so the distribution of metal fiber is most sparse. The degree of deformation on the tooth side is between the tooth root and tooth top, so the density of metal fiber tissue on the tooth side is also between the two.

### 2.2. Experimental Setup

#### 2.2.1. Extruded Tap and Workpiece

The taps used in the enterprise are M22 × 2 Wagner octagonal taps, which are composed of the working part and clamping part. The geometric parameters of the working part are obtained based on computer vision technology. The length of extrusion cone *l*_1_ is 8 mm, the length of correction part *l*_2_ is 18 mm, and the angle of extrusion cone *φ* is 4.5°, as shown in Figure 2. The workpiece used in the experiment is 42CrMO4 steel square with a round through hole. The size is 50 mm × 50 mm × 20 mm, and the diameter of the bottom hole is ∅21.15 mm, as shown in Figure 3.

#### 2.2.2. Measurement of Extrusion Torque and Temperature

The thread forming experiment is carried out on the horizontal machining center. The workpiece is placed vertically on the worktable and fixed between the pressing plate and worktable by tightening the nut. The tap is clamped by an automatic chuck and the machine tool spindle is used to drive the rotation of the tap so as to process the thread on the inner wall of the workpiece. The extrusion torque is measured by the SPIKE force measuring tool handle, and the extrusion temperature is measured by a high precision thermometer. In order to reduce the error of the measured temperature, it is necessary to make the thermocouple and the deformation area of the workpiece as close as possible, so a hole is drilled at 1/3 of the wall surface of the workpiece. The diameter of the hole is slightly larger than that of the thermocouple, which is 3.2 mm in diameter and 12.5 mm in depth. When measuring, the front end of the K-type thermocouple is inserted into the hole, and the other end is connected to a thermometer to record the data of extrusion temperature. The overall experiment is shown in Figure 4.

### 2.3. Experimental Results

#### 2.3.1. Extrusion Torque and Temperature

The experiment of the tapping process is carried out under the condition that the diameter of the bottom hole is ∅21.15 mm, the machine speed is 50 RPM, and the friction coefficient is 10# engine oil. The measured extrusion torque and temperature are shown in Figure 5 and Figure 6. It can be seen from figure that the extrusion torque and temperature increase at first and then decrease with the passage of time. The forming process of the internal thread can be divided into three main stages: Ⅰ extrusion stage, Ⅱ extrusion and correction stage, and Ⅲ correction stage. In Stage I, with the continuous extrusion of tapered teeth on the workpiece, the metal plastic deformation of the workpiece continues to occur. Due to the existence of friction force and friction torque in the extrusion process, the torque of the whole stage continues to increase from 0 N∙m. In Stage Ⅱ, there is both extrusion and correction effects, and the number of edges involved in extrusion and correction is increasing, so the extrusion torque is increased. Because there is no taper angle in the tap calibration part, the degree of metal plastic deformation of workpiece is smaller than that of Stage I, so the increase in extrusion torque is small at this stage. Furthermore, the maximum torque of the whole extrusion process also occurs in this stage; the maximum extrusion torque is 100.65 N∙m. In Stage Ⅲ, with the extrusion of the tap, the number of extrusion edges involved in the correction decreases continuously. When the tap is completely detached from workpiece, the extrusion torque drops to 0 N∙m. The extrusion temperature also increases with the increase in tap depth, and then decreases when the tap gradually withdraws from the workpiece. The highest extrusion temperature in this process is 57.37 °C.

#### 2.3.2. Thread Profile and Measurement of Threaded Tooth Height

The tooth height of thread will directly affect the connection strength of thread, so it is necessary to measure the thread height. The preparation process of the thread testing sample is as follows: first, a small piece of the sample is cut along the normal direction of thread at the through hole of the workpiece by wire cutting equipment. Then, the sample is cleaned and polished, as shown in Figure 7. The tooth height of the thread sample is measured by the shape measurement laser microscope system, as shown in Figure 8.

Figure 9 shows the morphology of the machined thread. It can be seen from the figure that there is a lack of meat at the tooth top. The forming mechanism of the thread in Section 2.1 explains that this is due to the fact that the metal of the workpiece is not fully filled with the groove of the tap under the extrusion of the tap. Although there is a lack of meat at the tooth top, the tooth height is 1.131 mm and the tooth height rate is 74.65%, which meets the requirements of thread connection. Therefore, under the condition of ensuring the thread connection strength, the appropriate gap between the tap and the workpiece can reduce the friction between the two. It can also reduce the torque and temperature in the machining process, so as to improve the forming quality of the thread.

#### 2.3.3. Analysis of Microstructure for Thread

The essence of extruded thread forming is the plastic deformation of the workpiece under the action of the tap; the metal material flows to the tap groove and fills gradually. In the thread forming process, when the metal is extruded by the tap, it will flow in the direction of the least resistance, so the degree of plastic deformation in different parts of the thread is also different. In order to study the microstructure of the thread, the threaded sample was eroded with an erosion solution suitable for 42CrMo4, that is, 4% nitric acid alcohol solution. The microstructure of the thread surface is observed by the shape measurement laser microscope system in Section 2.3.2. The microstructure of the thread is shown in Figure 10.

It can be seen from Figure 10a that the microstructure of the thread after extrusion has changed significantly. Compared with the part without extrusion, the degree of grain refinement and the density of fiber structure are significantly improved. When the thread is extruded by the tap, the metal of the workpiece flows in the groove between the two teeth of the tap. The grains are twisted and stretched along the direction of metal flow. With the increase in the amount for extrusion, the surface structure is compressed to fibrous. The method of extrusion does not block the flow of fiber tissue, and fiber tissue from the tooth root to tooth top is still continuous.

In the process of tap processing, the extrusion pressure on different parts of the thread is different, so the microstructure of tooth root, tooth side, and tooth top is different. The tooth root is directly under the action of the tapered tooth, so the tooth root is subjected to the greatest extrusion pressure, which results in the highest degree of plastic deformation and grain refinement in the tooth root. The metal fiber structure at the tooth root is pressed to such an extent that it is difficult to distinguish the grains, and it forms a curved streamline, as shown in Figure 10b. The extrusion pressure on the tooth side is less than that on the tooth root, so the plastic deformation and grain refinement are reduced. The density of metal fibers at the tooth side is also less than that at the tooth root, and the flow is similar to a straight line, as shown in Figure 10c. The extrusion pressure at the tooth top is the least, so the degree of plastic deformation and grain refinement is the lowest, and the metal fiber is relatively sparse, as shown in Figure 10d. When gradually moving away from the tooth root, the metal can hardly be affected by extrusion pressure, so the degree of plastic deformation and grain density do not increase. The metallographic structure is basically the same and there is no obvious deformation, showing the state of the original material, as shown in Figure 10e. Therefore, the processing by extrusion can improve the grain refinement and fiber structure density of metal materials, so as to improve the quality of the thread.

#### 2.3.4. Measurement of Hardness and Hardened Layer for Thread

As can be seen from the analysis of Section 2.3.3, the plastic deformation of the workpiece occurs under the extrusion of the tap, and the grains in the metal microstructure are elongated and fibrotic. In order to analyze the relationship between the microstructure and mechanical properties, the hardness and hardening layer of threads in different parts were measured by the automatic micro-Vickers hardness measurement system, as shown in Figure 11. The distance between measuring points is 0.05 mm. When measuring, the load of hardness tester is set to 200 g and the time to keep the load is set to 15 s. Because the tooth surface area of the internal thread is small and the shape is complex, the measuring point closest to the surface on the normal section of the internal thread is used as the evaluation basis of thread hardness. Figure 12 shows the hardness evaluation point at the tooth root.

The hardness changes of three parts along the layer depth are shown in Figure 13. As can be seen from the figure, there is a large difference in hardness between the tooth root, tooth top, and tooth side. The hardness at the tooth root is the highest, which is 358.87 HV_0.2_. The hardness at the tooth side is less than that at the tooth root, which is 337.45 HV_0.2_. The hardness at the tooth top is the smallest, which is 302.13 HV_0.2_. This can be seen from the analysis of the microstructure of the thread in Section 2.3.3. The degree of grain refinement and the density of metal fibers at the tooth root are greater than those at the tooth side. The degree of grain refinement and the density of metal fibers at the tooth side are greater than those at the tooth top. Therefore, the degree of grain refinement and the density of the rheological structure directly determine the strengthening degree of metal. The denser the rheological structure is, the higher the strengthening degree of metal is, and the hardness of metal will be improved. With the increase in the degree of plastic deformation, the depth of the hardened layer will also increase. The degree of plastic deformation at the tooth root is the largest, so the depth of the hardened layer is the deepest, which is 0.35 mm. The degree of plastic deformation at the tooth top is the smallest, so the depth of the hardened layer is the shallowest, which is 0.15 mm. Furthermore, the degree of plastic deformation on the tooth side is between the tooth root and tooth top, so the depth of the hardened layer is also between the two, at 0.2 mm.

From the hardness curves of three parts, it can be found that the hardness of the tooth root and tooth side decreases gradually with the depth of the layer. This is due to the fact that the plastic deformation mainly occurs on the surface of the thread. With the increase in the depth from the thread surface, the degree of plastic deformation decreases, which will lead to the decrease in work hardening, as shown by the change of hardness at the tooth root and tooth side. While the hardness at the tooth top increases at first and then decreases with the increase in layer depth. This is because the point at the tooth top is close to the lack of meat, and the metal material in the lack of meat is insufficient, so the point near this position has lower hardness.

## 3. Numerical Model and Validation

### 3.1. Establishment of Machining Model for Internal Thread

There are many combinations of different bottom hole diameter, machine tool speed, and lubricating medium used to improve the testing efficiency and avoid the production safety problems, such as tap fracture, caused by improper parameter selection. The influence of machining parameters on thread forming is analyzed by numerical simulation. According to the actual model of the tap and workpiece, the corresponding geometric model is established in 3D modeling software and imported into DEFORM-3D. The tap is set as a rigid body in the software, regardless of its deformation and wear [30,31]. The whole workpiece is meshed by the pre-processing function in the software. It is divided into 34,257 nodes and 143,657 elements, and the minimum grid size is 0.26. In order to obtain a detailed thread shape, the weight factor is set to 1 for local mesh refinement. Through the frame selection function of the refinement module, the refinement range is set to a cylinder with a height of 22 mm and a diameter of 25.5 mm, which covers the deformation zone of the workpiece. Furthermore, in order to improve the accuracy of the thread obtained by numerical simulation, the size ratio of the inside and outside of the frame selection is set to 10,000:1. After refinement, the minimum mesh size is reduced to 0.059, the number of nodes is increased to 115,007, and the number of elements is increased to 529,281. The position of the workpiece is fixed, and according to the relationship among the tap pitch, feed speed, and machine tool speed, the feed speed of the tap is set to 1.667 mm/s and the tap speed is 50 RPM. Using the shear friction model, the friction coefficient between the tap and workpiece is set to 0.12. Using the Newton-Raphson iterative algorithm, the simulation forced stop condition is set to the movement distance of the tap reaching 48 mm. The run step is set to 600 and the step size is set to 0.08. The finite element model of the workpiece and tap is shown in Figure 14.

### 3.2. Verification of Model

(1)Extrusion torque and temperature.

The curve of the extrusion torque obtained by numerical simulation and experiment is shown in Figure 15. As can be seen from the figure, the maximum extrusion torque obtained by experiment is 100.65 N∙m, while the maximum extrusion torque obtained by numerical simulation is 98.06 N∙m, with an error of only 2.57%. The maximum extrusion temperature measured by the thermometer is 57.37 °C, while the maximum extrusion temperature obtained by numerical simulation is 59.91 °C, with an error of 4.43%. This result verifies the accuracy of the numerical simulation.

(2)Thread profile.

In order to verify the accuracy of the thread obtained by numerical simulation, the numerical simulation of thread machining is carried out by using the parameters consistent with the experiment. The thread obtained by simulation is shown in Figure 16. Comparing Figure 16 with Figure 9, it can be found that the thread profile obtained by numerical simulation is similar to that obtained by experiment; both of them have the phenomenon of lack of meat at the tooth top. At the same time, the tooth height measured by the point tracking function in the numerical simulation is 1.162 mm and the experimental thread height is 1.131 mm. The error between the two is only 2.74%. This further verifies the reliability of the numerical simulation results.

## 4. Effect of Process Parameters on Thread Extrusion Process

### 4.1. Analysis of Bottom Hole Diameter

In order to determine the appropriate diameter of the prefabricated bottom hole, the numerical simulation was carried out under the condition that the diameter of the bottom hole was ∅21.05 mm, ∅21.10 mm, ∅21.15 mm, and ∅21.20 mm, respectively. Then, the optimal bottom hole diameter was selected according to the torque and temperature. The variation curves of extrusion torque and temperature under four diameters are shown in Figure 17 and Figure 18. It can be seen from the figure that with the decrease in the bottom hole diameter, the extrusion torque and temperature increase and the degree of lack of meat at the tooth top is decreasing. When the diameter of the bottom hole is ∅21.20 mm, the extrusion torque and temperature are lower because the tap needs to extrude less metal material. The maximum extrusion torque is 88.30 N·m and maximum extrusion temperature is 50.09 °C. At this time, the tooth top has an obvious lack of meat due to the insufficient amount of material flowing into the tapered tooth groove in the extrusion process. When the diameter of the bottom hole decreases to ∅21.05 mm, the amount of material extruded by the tap increases. So, the friction between the tap and workpiece increases sharply. The maximum extrusion torque increases to 118.72 N·m and maximum extrusion temperature increases to 72 °C. At this time, the lack of meat at the tooth top is small, which is due to the fact that a large amount of metal material flows into the grooves between tapered teeth during extrusion, and the excess metal material can gradually fill the grooves.

### 4.2. Analysis of Machine Tool Speed

In order to determine the appropriate machine tool speed, the numerical simulation was carried out under the condition that the machine tool speed was 30 RPM, 40 RPM, 50 RPM, and 60 RPM, respectively. The influence of rotation speed on extrusion torque and temperature was studied, and the optimal speed was selected. The variation curves of extrusion torque and temperature at four rotational speeds are shown in Figure 19 and Figure 20. It can be seen from the figure that the extrusion temperature increases with the increase in the machine tool speed, while the extrusion torque decreases first and then increases with the increase in rotational speed. When the machine speed is 30 RPM, the maximum extrusion torque is 92.17 N·m and the maximum extrusion temperature is 45.53 °C. When the machine speed increases to 40 RPM, the maximum extrusion torque decreases to 84.13 N·m, and the maximum extrusion temperature increases to 50.34 °C. When the machine speed increases to 60 RPM, the maximum extrusion torque increases to 109.59 N·m, and the maximum extrusion temperature increases to 57.25 °C. This is due to the fact that when the machine tool speed is in a high range, with the increase in the machine tool speed, the plastic deformation of the workpiece increases in unit time, and the friction between the workpiece and tap gradually increases. So, it will lead to the increase in extrusion temperature and extrusion torque.

### 4.3. Analysis of Lubricating Medium

In order to determine the suitable lubricating medium, choose from PDMS polydimethylsiloxane coolant, 10# engine oil, 20# engine oil, and 30# engine oil according to the extrusion torque and temperature. The different lubricating media in the numerical simulation are reflected in the different friction coefficients between the extruded tap and the workpiece. The setting of the friction coefficient in relation to different lubricating media in the numerical simulation is shown in Table 1. The variation curves of extrusion torque and temperature under four kinds of lubricating media are shown in Figure 21 and Figure 22. It can be seen from the figure that the greater the friction coefficient of lubricating medium used, the higher the torque and temperature in the extrusion process. When the lubricating medium is 30# engine oil, the maximum extrusion torque is 121.51 N·m and the maximum extrusion temperature is 71.49 °C. When the lubricating medium was replaced by PDMS polydimethylsiloxane coolant, the extrusion torque and temperature decreased due to the decrease in the friction coefficient, the maximum extrusion torque decreased to 88.42 N·m, and the maximum extrusion temperature decreased to 46.86 °C. In the experiment, it is shown that PDMS polydimethylsiloxane coolant reduces the direct contact between the edge teeth of the extruded tap and the metal material in the deformation zone of the workpiece. This is so that the relative slip shearing process between two surfaces is carried out in the interior of the lubricating layer. As a result, the energy consumption caused by friction in the extrusion process is greatly reduced, and then it reduces the torque and temperature.

## 5. Optimization Design

### 5.1. Design of Orthogonal Test Table

The key factors affecting the forming quality of the internal thread are extrusion torque and extrusion temperature. For the purpose of improving the quality of the thread, it starts by reducing the torque and temperature in the machining process. The selection of the bottom hole diameter, machine tool speed, and lubricating medium is optimized by the method of orthogonal test. According to the selected process parameters, a three-factor and four-level table was made; as shown in Table 2, a total of 16 experiments were carried out. According to the comprehensive balance method, the extrusion torque and extrusion temperature are analyzed, respectively, to obtain their respective optimal horizontal combination. Then, the optimized horizontal combinations under each index are synthesized, and finally the optimal horizontal combination is obtained. The orthogonal table designed according to the orthogonal test is shown in Table 3.

### 5.2. Data Analysis of Orthogonal Test

The range analysis obtained from the orthogonal test is shown in Table 4. According to the K value in the table, the effect curve of extrusion torque and extrusion temperature is drawn; this is also shown in Figure 23 and Figure 24. The figure shows the values of extrusion torque and temperature obtained when each factor takes different levels, which reflects the influence of various parameters on the torque and temperature in the forming process.

When taking extrusion torque as the optimization goal, it can be seen from Figure 23 that the extrusion torque in the thread machining process decreases gradually with the increase in the bottom hole diameter. When the diameter of the bottom hole is ∅21.20 mm, the extrusion torque in the extrusion process is at a minimum of 91.04 N·m. With the increase in the machine tool speed, the extrusion torque decreases at first and then increases. When the machine tool speed is 40 RPM, the extrusion torque is at a minimum of 96.95 N∙m. The extrusion torque increases with the increase in the friction coefficient of the lubricating medium. When the lubricant is PDMS polydimethylsiloxane coolant, the extrusion torque is at a minimum of 103.24 N∙m. According to the range analysis in Table 4, three factors have an influence on the extrusion torque in the forming process; the order of influence is: bottom hole diameter > machine tool speed > lubrication medium. Therefore, in order to obtain the minimum extrusion torque, the best processing parameters for the internal thread are: A4B2C1, that is, the diameter of bottom hole is ∅21.20 mm, the machine tool speed is 40 RPM, and the lubricating medium is PDMS polydimethylsiloxane coolant.

When taking extrusion temperature as the optimization goal, it can be seen from Figure 24 that the extrusion torque in the thread machining process decreases with the increase in bottom hole diameter. When the diameter of the bottom hole is ∅21.20 mm, the extrusion temperature is the lowest at 53.58 °C. The extrusion temperature increases gradually with the increase in machine tool speed. When the machine speed is 30 RPM, the extrusion temperature is the lowest at 60.25 °C. The extrusion temperature increases with the increase in the friction coefficient of the lubricating medium. When the lubricating medium is the PDMS polydimethylsiloxane coolant, the extrusion temperature is the lowest at 53.88 °C. According to the range analysis in Table 4, all three factors have an influence on the temperature in the thread forming process, and the influence order is as follows: bottom hole diameter > lubricating medium > machine tool speed. Therefore, in order to obtain the minimum extrusion temperature, the best processing parameters for the internal thread are as follows: A4B1C1, that is, the diameter of the bottom hole is ∅21.20 mm, the machine tool speed is 30 RPM, and the lubricating medium is PDMS polydimethylsiloxane coolant.

From the above analysis, we can see that when different results are taken as the optimization objectives, the optimization schemes obtained are not consistent. When extrusion torque and extrusion temperature are comprehensively considered, the diameter of the bottom hole and lubricating medium are the same. The diameter of the bottom hole is ∅21.20 mm and lubricating medium is PDMS polydimethylsiloxane. However, the selection of the machine tool speed is different. When extrusion torque is taken as the optimization index, the machine tool speed is selected to be 40 RPM, and the extrusion temperature is 60.91 °C. When extrusion temperature is taken as the optimization index, the machine tool speed is chosen as 30 RPM, and the extrusion temperature is 60.25 °C. The temperature difference between two schemes is only 0.66 °C. It can be found from Figure 24 that extrusion temperature increases with the acceleration of machine tool speed. However, when the rotational speed is at 30~50 RPM, the increase in extrusion temperature is small, which has little effect on the extrusion temperature. Furthermore, the extrusion temperature is not very high in the forming process and the influence of extrusion torque on the life of the tap and the quality of the thread is much higher than the extrusion temperature, so the machine speed of 40 RPM is chosen. Based on the above analysis, the final optimal scheme is A4B2C1; that is, the diameter of the bottom hole is ∅21.20 mm, the machine tool speed is 40 RPM, and the lubricating medium is PDMS polydimethylsiloxane coolant.

### 5.3. Verification of Optimization Results

#### 5.3.1. Extrusion Torque and Temperature

According to the optimized parameters obtained from the above analysis, the experiment for internal thread forming is carried out. The variation curves of torque and temperature before and after optimization are shown in Figure 25 and Figure 26, respectively. As can be seen from the figure, when the thread is extruded using the optimized processing parameters, the maximum extrusion torque is 92.01 N∙m and the maximum extrusion temperature is 51.96 °C. While the optimized parameters are used to extrude the thread, the maximum extrusion torque is 74.28 N∙m and the maximum extrusion temperature is 44.13 °C. Compared with that before optimization, the maximum extrusion torque after optimization is reduced by 19.27%, and the maximum extrusion temperature is reduced by 15.07%. The results show that the optimized processing parameters can greatly reduce the extrusion torque and temperature, thus effectively improving the forming quality of the thread and the service life of the tap.

#### 5.3.2. Threaded Tooth Height

Figure 27 shows the connection strength of the thread before and after optimization. It can be seen that the tooth height of the thread before optimization is 1.131 mm, the tooth height rate is 74.65%, and thread connection strength is 99.39%. While the tooth height of the optimized thread is 1.079 mm, the tooth height rate is 71.22, and the thread connection strength is 98.77%. From the relationship between tooth height rate and thread connection strength, we can see that when the thread height rate reaches 70%, the thread connection strength is close to 100%. At this time, if we continue to increase the tooth height ratio of the thread, although the connection strength still improves, the room for improvement is limited. When the thread connection strength reaches 100%, and one continues to increase the tooth height, the connection strength is no longer improved. So, on the premise of satisfying the thread connection strength, it is not necessary to pursue the tooth height rate excessively. Otherwise, the too-small diameter of the bottom hole will aggravate the friction between the tap and workpiece, which will greatly increase the wear of the tap and reduce its life. Therefore, using the optimized parameters to extrude the thread can not only ensure the strength of thread connection but also reduce the wear of the tap and workpiece and prolong their service life.

#### 5.3.3. Hardness and Hardened Layer for Thread

The hardness changes of three parts along the layer depth after optimization are shown in Figure 28. As can be seen from the figure, the hardness of the optimized tooth root is 365.52 HV_0.2_ and the depth of the hardened layer is 0.35 mm. The hardness of the optimized tooth top is 305.50 HV_0.2_ and the depth of the hardened layer is 0.2 mm. The hardness of the optimized tooth side is 341.40 HV_0.2_ and the depth of the hardened layer is 0.25 mm. While it can be found from Figure 13 that the hardness of the tooth root before optimization is 358.87 HV_0.2_, the depth of the hardened layer is 0.3 mm. The hardness of the tooth top before optimization is 302.13 HV_0.2_, and the depth of the hardened layer is 0.15 mm. The hardness of the tooth side before optimization is 337.45 HV_0.2_ and the depth of the hardened layer is 0.2 mm. By comparing the thread hardness before and after optimization, it can be found that the optimized thread hardness increases by about 5 HV_0.2_ and the hardened layer depth increases by about 0.05 mm. The results show that the optimized processing parameters can effectively improve the forming quality of thread.

## 6. Conclusions

In order to improve the forming quality of thread, the effects of bottom hole diameter, machine tool speed, and lubricating medium on extrusion torque and temperature are studied by numerical simulation. For extrusion torque, the influence order is bottom hole diameter > machine tool speed > lubricating medium. For extrusion temperature, the influence order is bottom hole diameter > machine tool speed > lubricating medium. Through orthogonal optimization, it is determined that the bottom hole diameter should choose ∅21.20 mm, machine tool speed should choose 40 RPM, and lubricating medium should choose PDMS polydimethylsiloxane coolant. When the optimized parameters are selected for processing, the maximum extrusion torque and the maximum extrusion temperature are reduced by 19.27% and 15.07%, respectively. The hardness of the optimized thread increases about 5 HV_0.2_ and the depth of the hardened layer increases about 0.05 mm. This proves the effectiveness of the optimized process.

Compared with the existing research, the method of combining numerical simulation and experiment is used to determine the best process parameters for machining the M22 × 2 thread. With the help of numerical simulation, the processing cost can be reduced and the processing safety problems, such as tap fracture caused by improper parameter selection, can be avoided. The accuracy of the finite element model and the reliability of the optimization results are verified by experiment. This can prevent the limitations of numerical simulation, such as the thread morphology not being able to be observed in detail and the hardness not being able to be measured. This method is also suitable for determining the parameters of other screw threads. However, there are some unstable factors affecting thread quality in actual machining, which cause a certain error between actual machining results and numerical simulation results. In the future, numerical simulation of thread processing, the elastic recovery of the workpiece, and the wear of the tap should be taken into account, so as to make the numerical simulation of thread forming closer to the actual.

## Figures and Tables

**Figure 1 materials-15-03160-f001:**
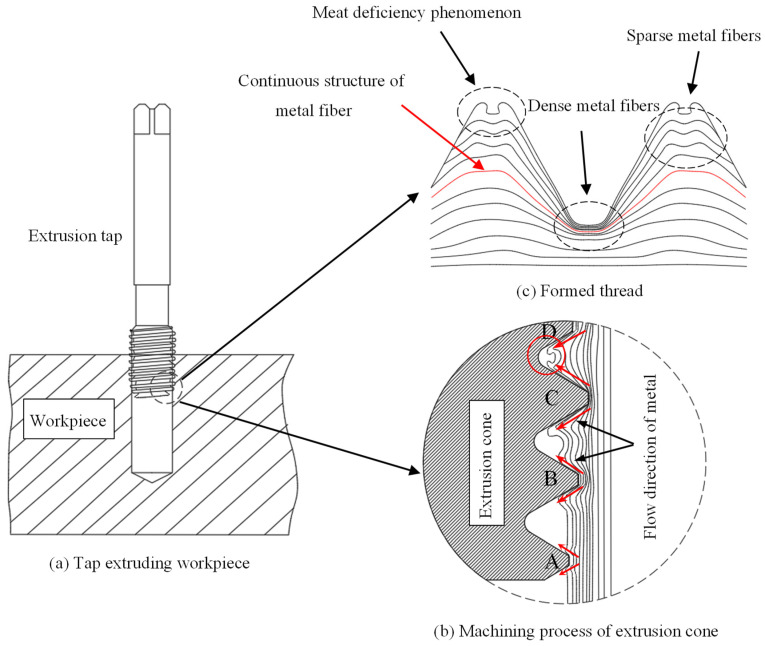
Schematic diagram of thread machining process.

**Figure 2 materials-15-03160-f002:**
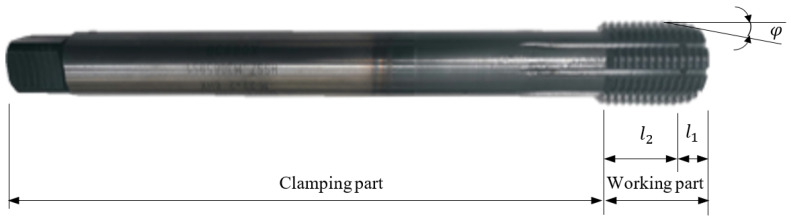
Structure of extrusion tap.

**Figure 3 materials-15-03160-f003:**
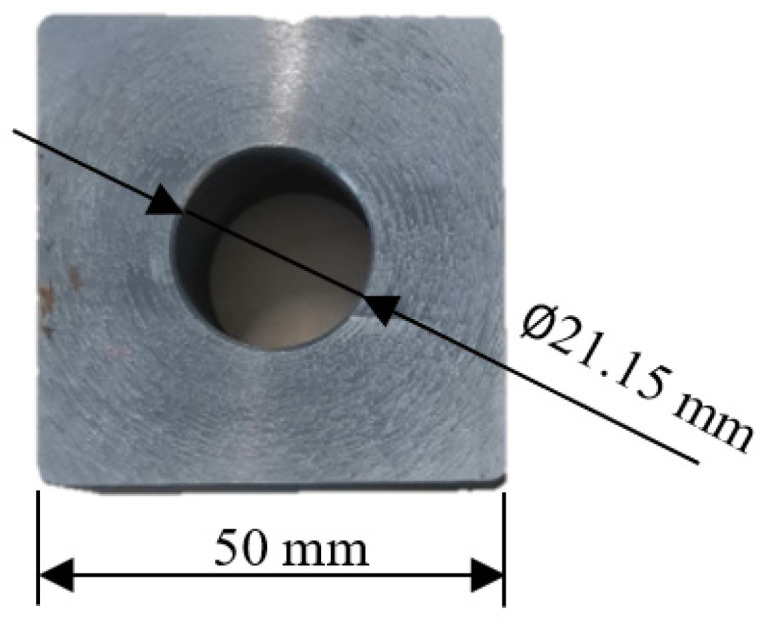
Structure of workpiece.

**Figure 4 materials-15-03160-f004:**
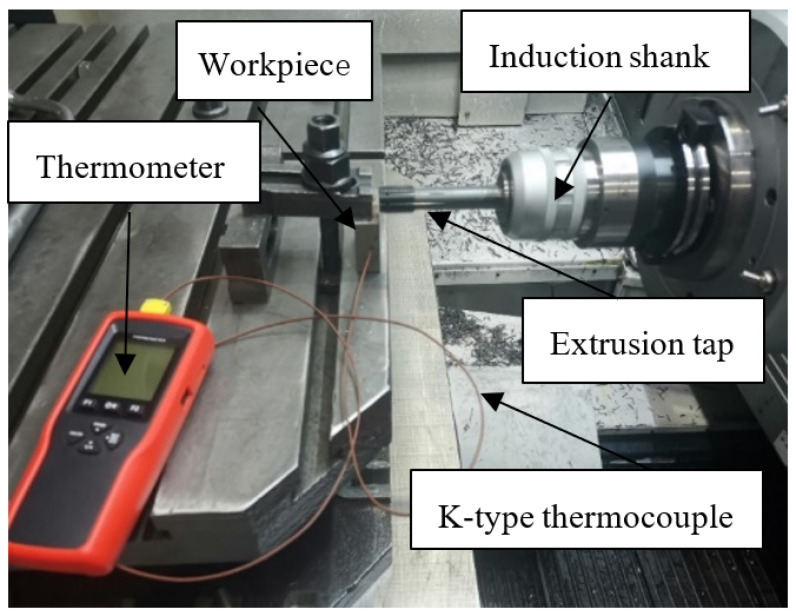
Experiment of thread machining.

**Figure 5 materials-15-03160-f005:**
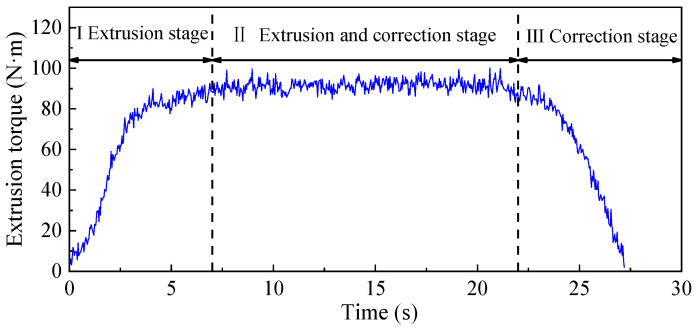
Extrusion torque.

**Figure 6 materials-15-03160-f006:**
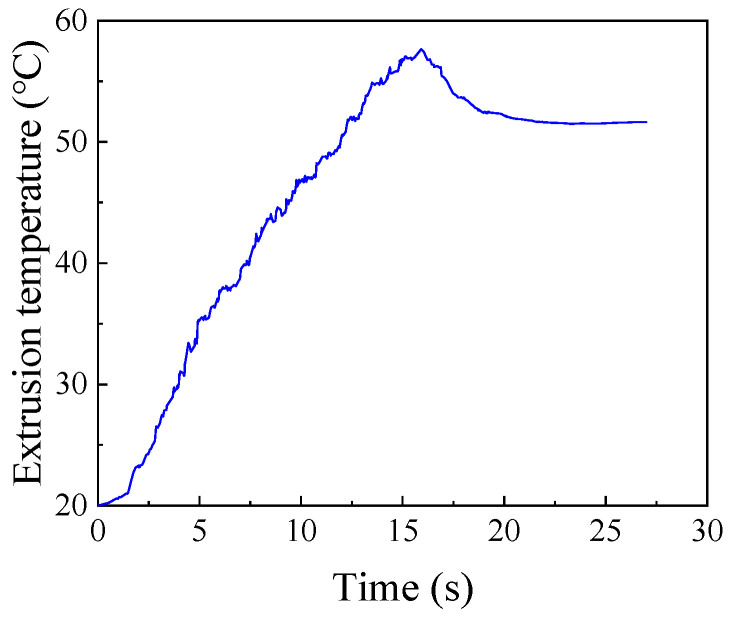
Extrusion temperature.

**Figure 7 materials-15-03160-f007:**
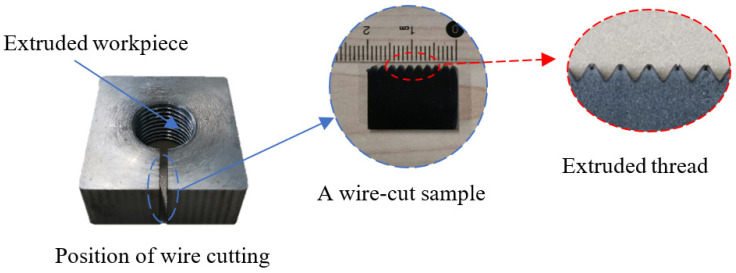
Thread sample to be measured.

**Figure 8 materials-15-03160-f008:**
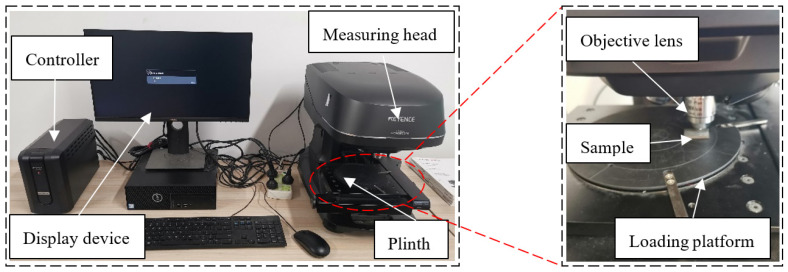
Laser microscopic system for shape measurement.

**Figure 9 materials-15-03160-f009:**
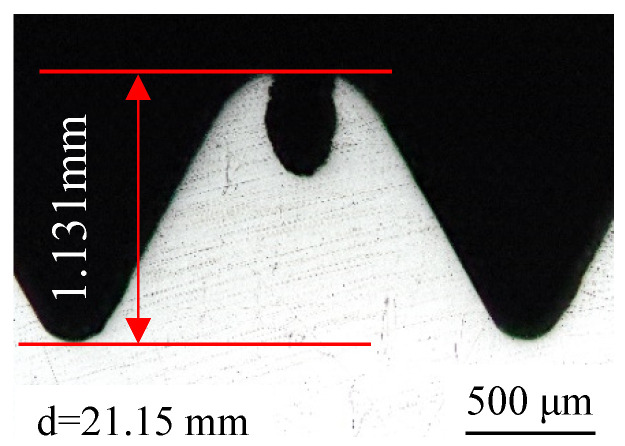
Morphology of machined thread.

**Figure 10 materials-15-03160-f010:**
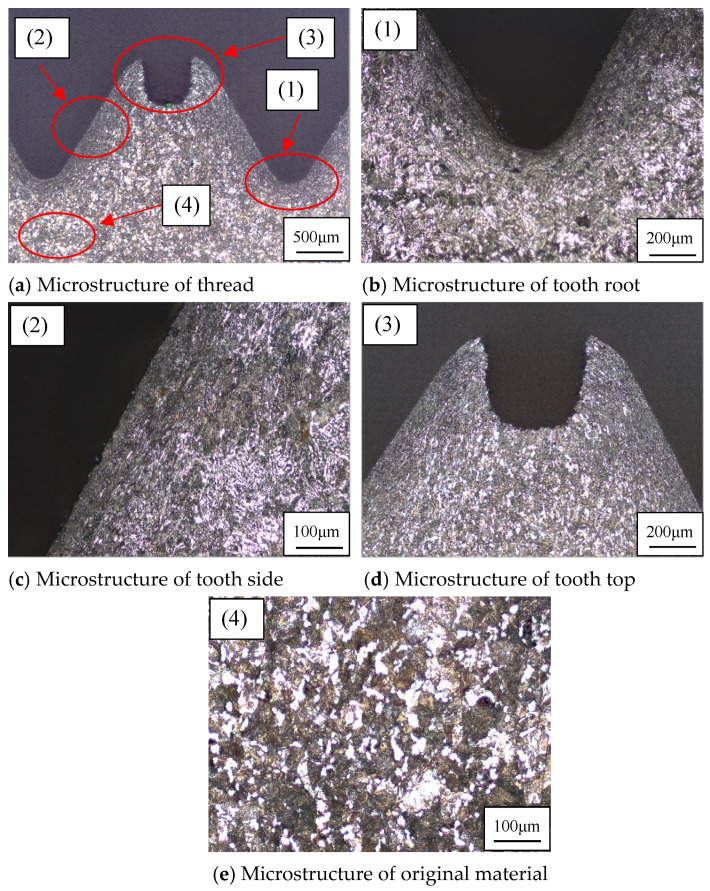
Microstructure of extruded thread.

**Figure 11 materials-15-03160-f011:**
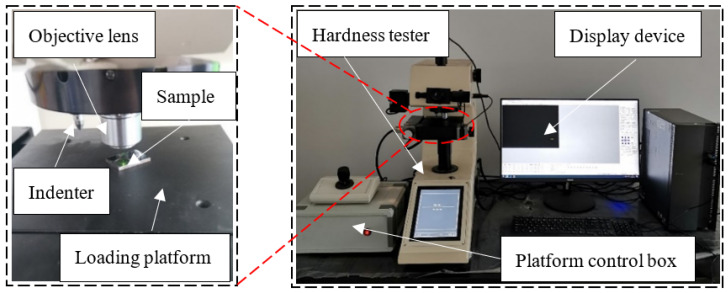
Automatic micro-Vickers hardness measuring system.

**Figure 12 materials-15-03160-f012:**
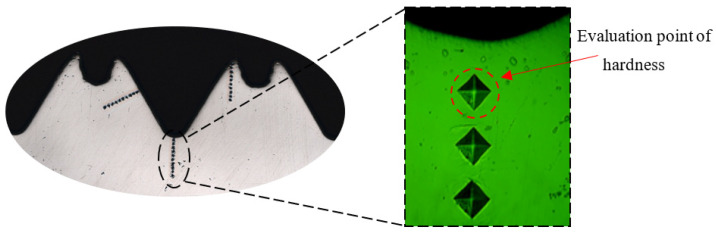
Hardness evaluation point of thread root.

**Figure 13 materials-15-03160-f013:**
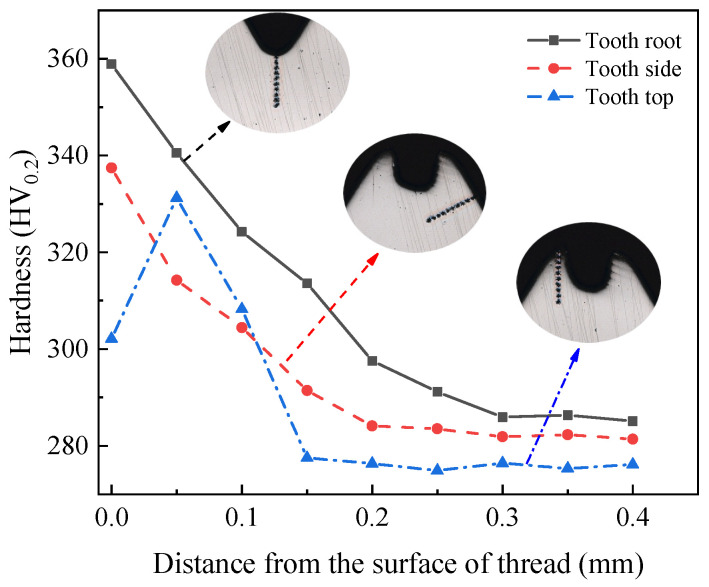
Measurement of thread hardness.

**Figure 14 materials-15-03160-f014:**
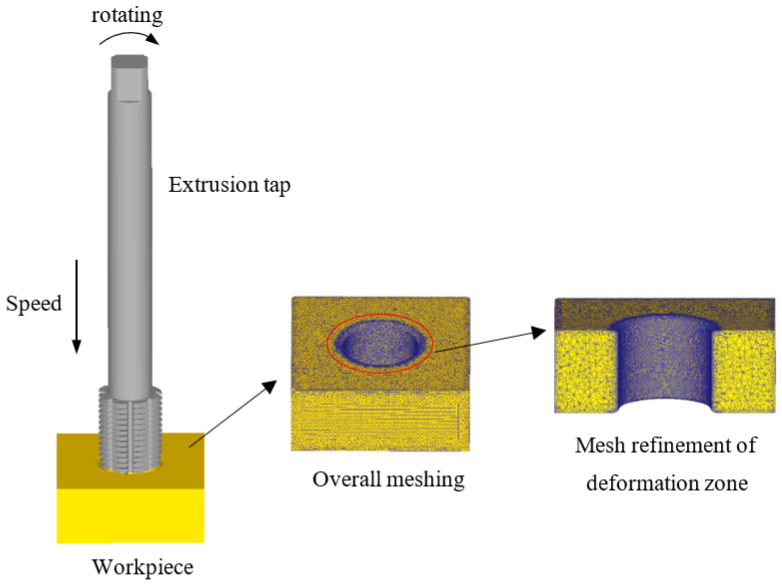
Finite element model of workpiece and tap.

**Figure 15 materials-15-03160-f015:**
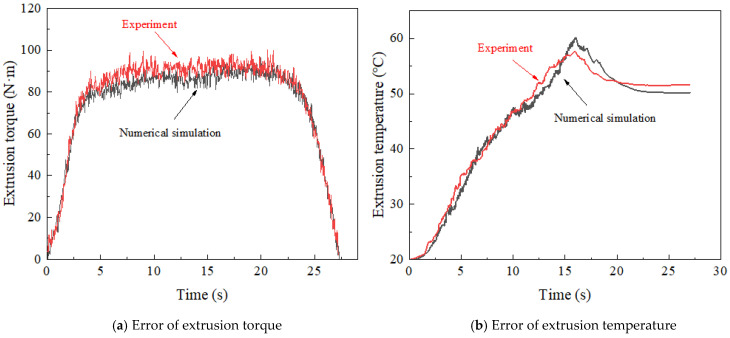
Error between numerical simulation and experiment.

**Figure 16 materials-15-03160-f016:**
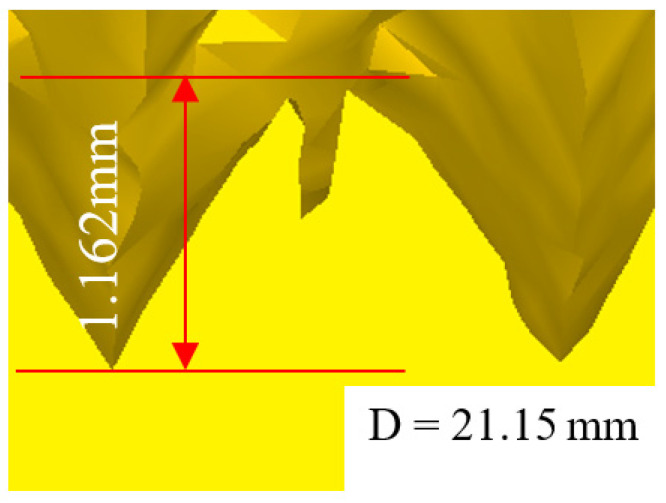
Thread morphology in numerical simulation.

**Figure 17 materials-15-03160-f017:**
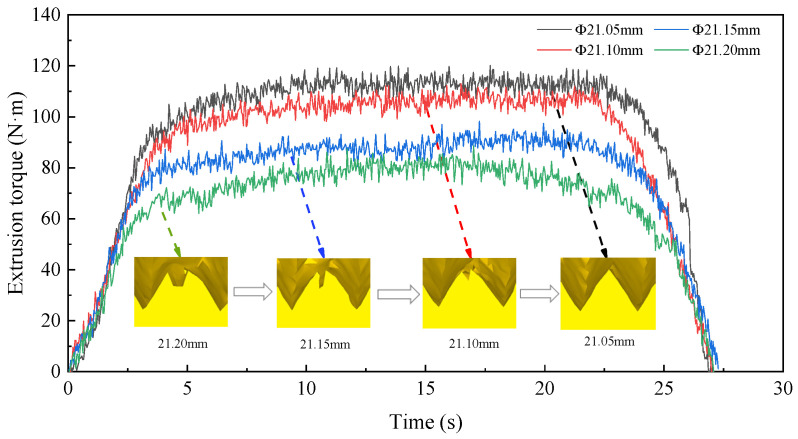
Influence of bottom hole diameter on torque.

**Figure 18 materials-15-03160-f018:**
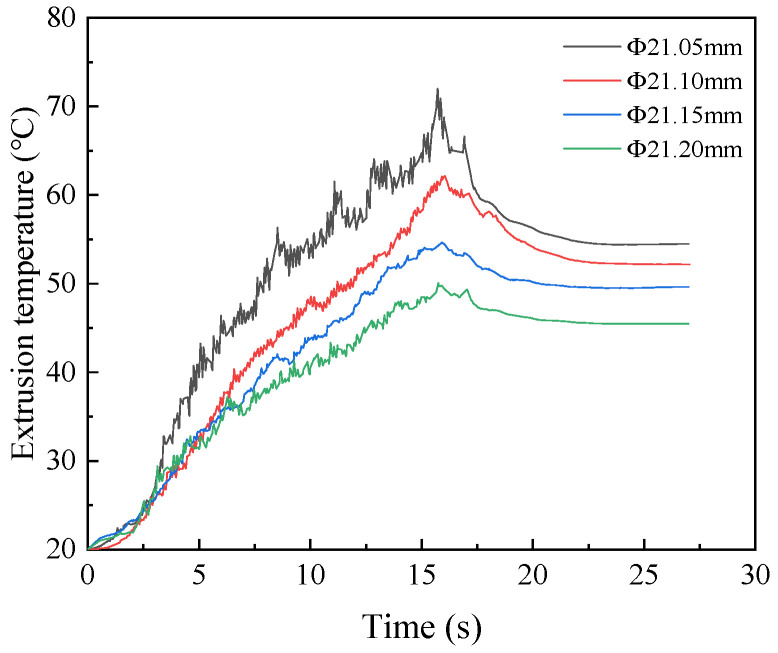
Influence of bottom hole diameter on temperature.

**Figure 19 materials-15-03160-f019:**
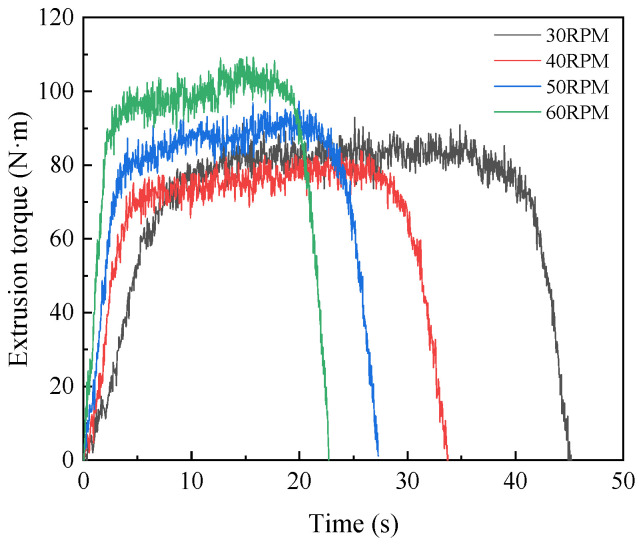
Influence of machine tool speed on torque.

**Figure 20 materials-15-03160-f020:**
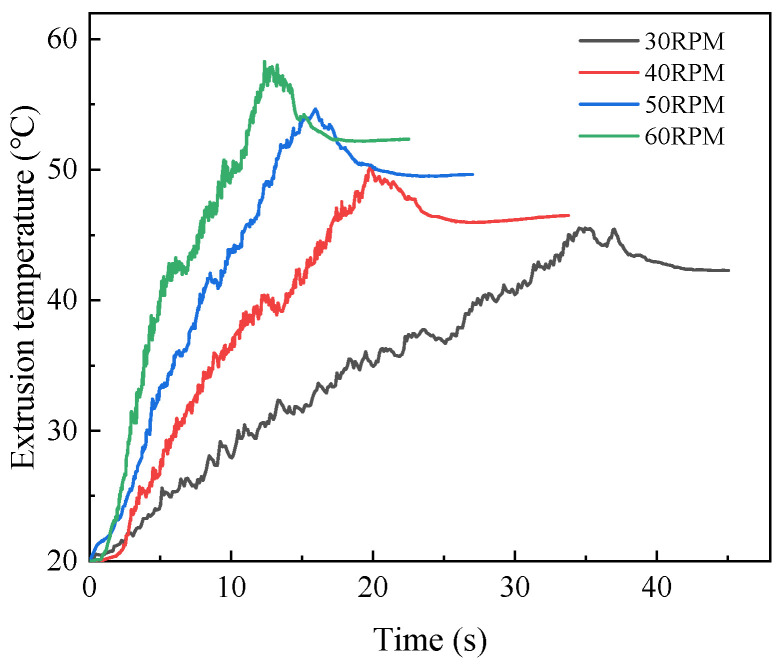
Influence of machine tool speed on temperature.

**Figure 21 materials-15-03160-f021:**
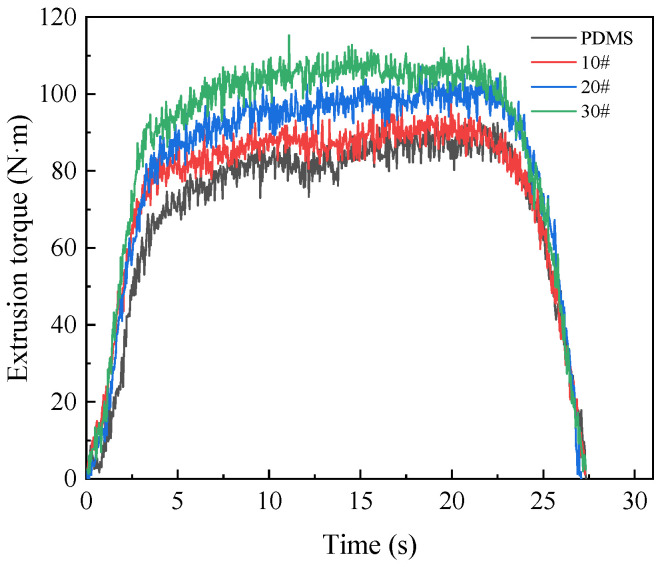
Influence of lubricating medium on torque.

**Figure 22 materials-15-03160-f022:**
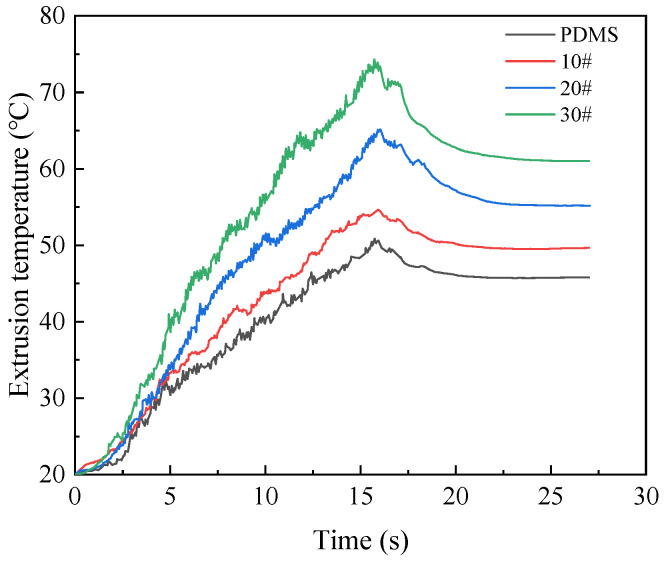
Influence of lubricating medium on temperature.

**Figure 23 materials-15-03160-f023:**
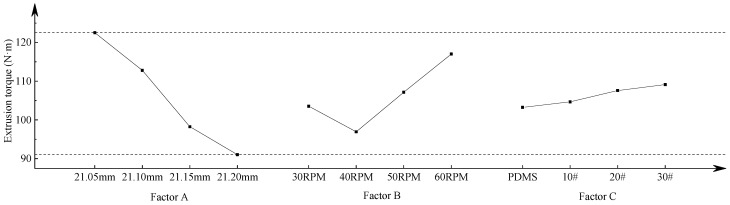
Effect curve of extrusion torque.

**Figure 24 materials-15-03160-f024:**
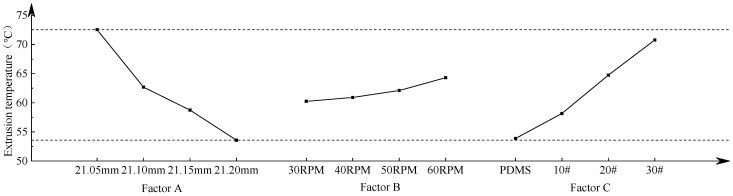
Effect curve of extrusion temperature.

**Figure 25 materials-15-03160-f025:**
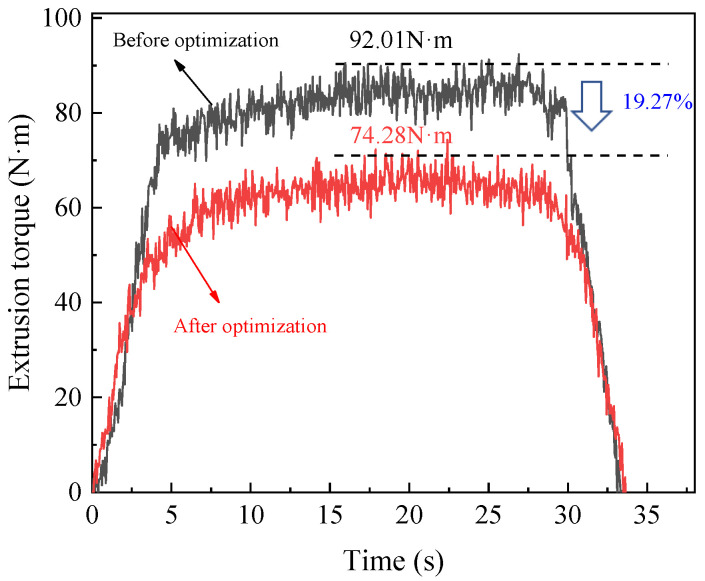
Optimization results of extrusion torque.

**Figure 26 materials-15-03160-f026:**
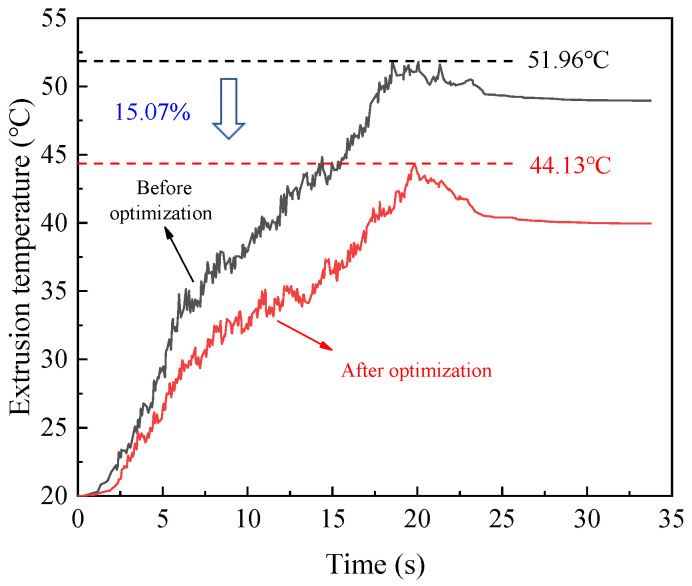
Optimization results of extrusion temperature.

**Figure 27 materials-15-03160-f027:**
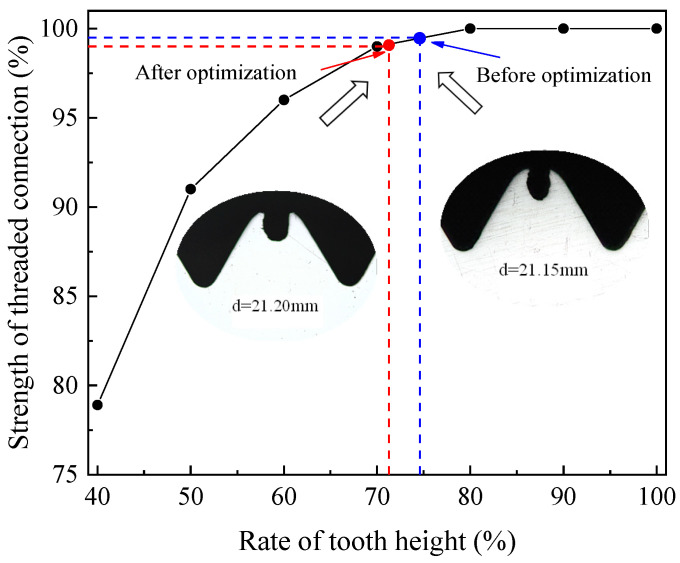
Strength of threaded connection before and after optimization.

**Figure 28 materials-15-03160-f028:**
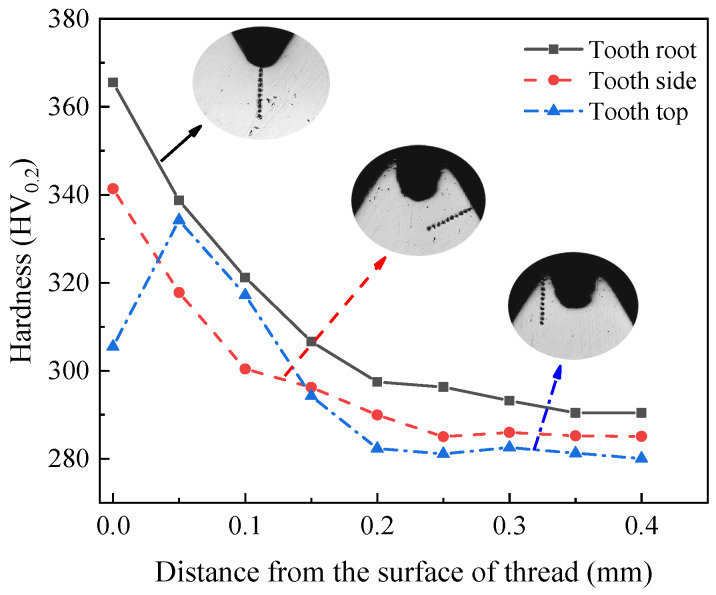
Measurement of thread hardness after optimization.

**Table 1 materials-15-03160-t001:** Viscosity of different lubricating media and friction coefficients in numerical simulation.

Lubricating Medium	Kinematic Viscosity at 40 °C (mm^2^/s)	Friction Coefficient in Numerical Simulation
PDMS polydimethylsiloxane coolant	10	0.08
10# engine oil	12	0.12
20# engine oil	23	0.2
30# engine oil	32	0.25

**Table 2 materials-15-03160-t002:** Factor level.

Level	A Diameter of Bottom Hole (mm)	B Machine Tool Speed (RPM)	C Lubricating Medium
1	21.05	30	PDMS polydimethylsiloxane coolant
2	21.10	40	10# engine oil
3	21.15	50	20# engine oil
4	21.20	60	30# engine oil

**Table 3 materials-15-03160-t003:** Orthogonal design of processing parameters.

Number	Diameter of Bottom Hole (mm)	Machine Tool Speed (RPM)	Lubricating Medium	Torque (N·m)	Temperature (°C)
1	21.05	30	PDMS	115.79	62.25
2	21.05	40	10#	113.56	67.59
3	21.05	50	20#	126.58	77.15
4	21.05	60	30#	134.13	83.17
5	21.10	30	10#	107.28	57.72
6	21.10	40	PDMS	101.57	54.17
7	21.10	50	30#	119.24	70.28
8	21.10	60	20#	123.14	68.55
9	21.15	30	20#	98.52	59.12
10	21.15	40	30#	90.54	67.74
11	21.15	50	PDMS	94.42	50.86
12	21.15	60	10#	109.59	57.25
13	21.20	30	30#	92.56	61.89
14	21.20	40	20#	82.13	54.12
15	21.20	50	10#	88.30	50.09
16	21.20	60	PDMS	101.17	48.23

**Table 4 materials-15-03160-t004:** Range analysis.

Test Index	Factors	K1	K2	K3	K4	Range Value R
Extrusion torque (N·m)	A	122.52	112.81	98.27	91.04	31.48
	B	103.54	96.95	107.14	117.01	20.06
	C	103.24	104.68	107.59	109.12	5.88
Extrusion temperature (°C)	A	72.54	62.68	58.74	53.58	18.96
	B	60.25	60.91	62.10	64.30	4.06
	C	53.88	58.16	64.74	70.77	16.89

## Data Availability

Not applicable.

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
