# Peer review of "Research and Optimization of Process Parameters for Internal Thread Forming Based on Numerical Simulation and Experimental Analysis"

_materials, 2022, doi:10.3390/ma15093160_

Round 1
Reviewer 1 Report
In this paper, the authors investigate the forming quality of extruded thread through finite element analysis and experiment method. The optimized results are obtained to reduce the influence of the extrusion torque and temperature on the thread quality.
To further improve the presentation, the following amendments are recommended:
- There are many writing mistakes in the Introduction, for example, (1) is the first word in the Title “Title Research and optimization of process parameters of cold extrusion for internal thread based on numerical simulation and experimental analysis” required? (2) The format of references 1-20 is different from the following, and there is no reference 21. (3) "Error! Reference source not found" appears many times in the introduction.
Therefore, please check the full paper without any writing errors
- In Section 3.1 finite element modeling, the description of mesh information need to be added, including the details of local mesh subdivision, which has a certain impact on the simulation accuracy.
- The authors should include in the paper possible advantages and disadvantages of the used method, compared to existing methods.
4. Would be possible to add more evidences about the amount of error between the results of numerical simulation and experiment?
Reviewer 2 Report
The manuscript presents a state of the art study. The research is well structured however the text needs to be revised heavily before publication.
The text needs to be proofread as there are several grammatical errors in the manuscript. The process studies is thread forming not cold extrusion.
There are a few occurrences of reference not found in the literature review.
The study needs to be restructured as in the current state results of the model are mixed with experimental ones.
The units of measurement must be consistent with the commonly used terms in literature. The term machine tool speed is measured in the manuscript in r/min. It must be referred to as RPM.
The location and distance of the pyrometer must be specified and compared with the results from deform. How were the temperature readings obtained in the trials with lubricant?
Regarding the properties of the engine oils how were the values for kinematic viscosity and friction coefficient obtained?
In the optimisation trial results, the torque and temperature are reduced but the time for machining the feature increased. this is expected as the machine speed was decreased. In reality, this is expected and could not be considered optimisation. was there any consideration of multi-objective optimisation?
In figure 15 the results from the machining trials for the optimised scenario have a significant difference to the simulated ones, how can this be explained?
